# Total Knee Arthroplasty in Patients with Primary Sjögren’s Syndrome: A Retrospective Case-Control Study Matched Patients without Rheumatic Diseases

**DOI:** 10.3390/jcm11247438

**Published:** 2022-12-15

**Authors:** Songlin Li, Xi Chen, Ruichen Ma, Shanni Li, Hongjun Xu, Jin Lin, Xisheng Weng, Wenwei Qian

**Affiliations:** 1Department of Orthopedic Surgery, Peking Union Medical College Hospital, Chinese Academy of Medical Sciences and Peking Union Medical College, Beijing 100006, China; 2Department of Orthopedic Surgery, West China Hospital, Sichuan University, Chengdu 610017, China; 3School of Medicine, Tsinghua University, Beijing 100190, China

**Keywords:** primary Sjögren’s syndrome, TKA, osteoarthritis, outcomes, quality of life

## Abstract

Background: The number of patients with primary Sjögren’s syndrome (pSS) who require total knee arthroplasty (TKA) is expected to increase, and there are few studies describing their outcomes. This research was focused on the evaluation of a TKA cohort in pSS patients and to compare outcomes with those of matched individuals from the general population. Methods: From 2004 to 2020, we found 36 TKAs in 30 patients with pSS from the single-institution arthroplasty registry, and they were matched for age, gender, bilateral or unilateral surgery, American Society of Anesthesiologists (ASA) score, and year of surgery with 72 TKAs in 60 osteoarthritis patients without rheumatic diseases (1:2 ratio). Perioperative outcomes were obtained, and clinical evaluations were performed at the last follow-up. Results: After a mean six-year follow-up, both cohorts had similar knee function and health-related quality of life outcomes. The pSS group had more patients with post-operative anemia and hypoalbuminemia and more patients needing platelet transfusion. There were no significant differences in other complications, the rates of 90-day readmission, or overall revision. By multivariate analysis, the influencing factor for anemia in pSS patients was lower preoperative hemoglobin (OR = 0.334, 95% CI (0.125–0.889), *p* < 0.05). Conclusions: Our study demonstrated that pSS patients who received TKA could achieve comparable clinical outcomes to the general population. However, more attention should be paid to the perioperative hematological management of pSS patients who underwent TKA.

## 1. Introduction

Primary Sjögren’s syndrome (pSS) is a chronic autoimmune inflammatory disease characterized by a reduction in lacrimal and salivary gland function along with consequent dry eyes and mouth as well as occurring independent of other autoimmune diseases [1,2,3]. In addition, there may be a variety of other disease manifestations involving multiple organs and organ systems. With standardization of the use of glucocorticoids and immunosuppressants, pSS is considered to be a chronic, slow-developing, non-life-threatening disease for the vast majority of patients, with a 10-year survival rate of more than 90%. Due to the disease’s complexity, patients with pSS typically require several different medical specialists to ensure that they are appropriately treated.

About 50% of patients with pSS report arthritis or joint pain [4], and this most commonly occurs in the knee joints of the lower extremities [5,6]. In the later stages of knee arthritis, total knee arthroplasty (TKA) is considered the definitive treatment [7,8]. For other rheumatic diseases such as rheumatoid arthritis (RA) and systemic lupus erythematosus (SLE), TKA also significantly improves knee function and quality of life in these patients, although complications and long-term survival are controversial [9,10,11,12]. However, the clinical outcome of TKA in pSS patients is lacking. Prior research has suggested that Sjögren’s syndrome (SS) patients have a higher rate of blood transfusion after TKA [13]. However, research has not distinguished between pSS and secondary Sjögren’s syndrome (sSS), and there has been limited research on clinical outcomes.

The number of patients undergoing TKA is expected to continue to increase as the number of pSS survivors increases and as it improves patients’ quality of life. Despite this increased need, practitioners may be hesitant to proceed with TKA in this patient population due to the increased risk of potential peri-operative complications associated with multi-organ involvement and the administration of glucocorticoids and immunosuppressants. In this research, we assessed a large single-center database to investigate if it is possible for pSS patients undergoing TKA to achieve similar clinical outcomes and complications by case-matching patients with knee osteoarthritis who had no history of the rheumatic diseases. Particularly, we assessed (1) knee function, (2) quality of life, (3) blood loss and transfusion, and (4) complications.

## 2. Materials and Methods

### 2.1. Study Population

A retrospective study was carried out in which patients diagnosed with pSS and knee osteoarthritis were screened from the knee arthroplasty database. Each medical record was reviewed to exclude patients suffering from other rheumatic diseases or who had undergone revision surgery or emergency surgery for fractures. From January 2004 to December 2020, a total of 34 pSS patients underwent TKA for knee osteoarthritis, and four of the patients could not be reached and contacted during follow-up. The final study cohort included 36 TKAs in 30 patients, all of whom were women, and 6 of whom had received simultaneous bilateral TKAs. All patients met the 2002 or 2016 diagnostic criteria for pSS [14,15]. The diagnostic records of rheumatologists in our hospital were recorded in the medical record system. Before surgery, these patients also received evaluation of disease activity and perioperative management recommendations from rheumatologists. The mean follow-up time was 6.84 ± 4.69 years. The mean body mass index (BMI) before surgery was 24.15 ± 2.48 kg/m^2^ (range 20.9–30.2). Average age at diagnosis of pSS was 55.87 ± 7.81 years (range 34–67), average age at surgery was 63.33 ± 8.28 years (range 40–77), and average time from diagnosis of pSS to surgery was 7.47 ± 4.79 years (range 2–23). See Appendix A for details.

To reduce selection bias and potential confounding effects, a matched comparison cohort was subsequently made as the non-pSS group. In the database of 6623 TKAs, we identified a comparison cohort at a ratio of 1:2. The cohorts were matched for age (within one year), gender, bilateral or unilateral surgery, American Society of Anesthesiologists (ASA) scores (within one score), and surgical data (within one year). If more than two patients in the non-pSS group were matched with the corresponding pSS patients, the two patients with the smallest difference in surgical date were selected as control patients. Sixty patients who underwent primary TKA for knee osteoarthritis without other rheumatic diseases were enrolled. Twelve patients underwent simultaneous bilateral surgery. Figure 1 shows the study flow. All patients obtained informed consent before inclusion, and the study was approved by the institutional review board (IRB) of Peking Union Medical College Hospital (I-22PJ463).

All surgeries were performed by the same group of high-volume surgeons with the same approaches and implants of the same designs. Perioperative analgesia, infection prevention, and postoperative rehabilitation were similar. In addition, both groups received standard thromboprophylaxis regimens in accordance with international guidelines and recommendations at the time of surgery. Routine follow-up was 1, 3, 6, 12 months, and then annually. Outpatient medical records are available for all patients.

### 2.2. Data Collection

Demographic information was obtained from the electronic medical record system, including age, sex, BMI, ASA(American Society of Anesthesiologists) scores, operation time, and hospital stay. Age-adjusted Charlson Comorbidity Index(aCCI) was calculated excluding the diagnosis of pSS. Alignment information was obtained from medical records or preoperative imaging data.

For patients in the pSS cohort, age at pSS diagnosis, time from pSS diagnosis to surgery, organ accumulation, laboratory indicators, and treatment status were also recorded. pSS disease activity was assessed using EULAR Sjögren’s Syndrome Patient Reported Index (ESSPRI), which included three major categories (dryness, fatigue, and pain (including joint and/or muscle pain)) measured on a digital scale of 0 to 10 using a visual analog score (VAS) [16]. The weights of the three categories were consistent, and the final score was the average of the scores of the three categories. Diseases involving lung (interstitial lung disease, multiple pulmonary bullae, pulmonary nodular amyloidosis, etc.), kidney (renal tubular acidosis, renal tubular injury, renal interstitial lesions, etc.), digestive system (primary biliary cholangitis, autoimmune hepatitis, autoimmune sclerosing pancreatitis, etc.), nervous system (small fiber neuropathy, most mononeuropathy, demyelinating disease, etc.), hematologic system (anemia, autoimmune hemolytic anemia, thrombocytopenia, monoclonal gamma globulinsis, lymphoma, etc.), and autoimmune thyroid disease were also recorded. Hematologic markers such as anti-Sjogren’s Syndrome A(anti-SSA), anti-Sjogren’s Syndrome B(anti-SSB), antinuclear antibody (ANA), and rheumatoid factor (RF) for diagnosis were recorded. Complement 3(C3), complement 4(C4), and IgG levels used to assess disease activity were also recorded. Preoperative use of glucocorticoids, hydroxychloroquine, total glucosides of paeony, and disease-modifying anti-rheumatic drugs (DMARDs) were recorded.

The Western Ontario and McMaster Universities Osteoarthritis Index (WOMAC), a patient-reported outcome measure (PROM), was used to evaluate knee function [17]. It includes pain, stiffness, and joint function, and consists of 24 items, including basic signs and symptoms of osteoarthritis. Each item is scored from zero to four, with zero indicating normal knee function and four indicating maximum knee dysfunction. The total WOMAC score is the sum of the scores, and the higher the score, the more severe the symptoms. The pain degree after activity was evaluated by VAS score [18]. The method entails the use of a ruler about 10 cm long, with 10 scales on one side and “0” and “10” at both ends, respectively. The Forgotten Joint Score (FJS) is an effective tool for assessing the postoperative satisfaction of patients undergoing TKA [19]. It is designed to test the degree of “forgetting” of the prosthesis in daily life. It is a 12-question questionnaire that combines variables such as pain, stiffness, activities of daily living, patient expectations, patient activity level, and psychosocial factors. The total score is 100, and the higher the score, the more the patient forgot about the artificial joint.

EuroQol 5-Dimensions (EQ-5D) and EuroQol-visual analogue scales (EQ-VAS) were used to assess the quality of life of patients [20,21]. The five dimensions are mobility, anxiety/depression, pain/discomfort, usual activities, and self-care. EQ-VAS is the subjective score of respondents’ current health status, which ranges from 0 to 100 points. The higher the score, the better the self-health status. The concept of these evaluation indicators and the meaning of different values was explained to patients in detail before they performed the patient-reported outcome measures. At the last follow-up, patients’ satisfaction with the surgery was recorded on a scale of 100, with 0 for no satisfaction at all and 100 for extreme satisfaction. Satisfaction was assessed anonymously.

Patient hematologic indicators were collected, including preoperative hemoglobin (Hb), albumin, plasma potassium, and platelet count, and postoperative minimum values of hemoglobin, albumin, plasma potassium, and platelet count. Postoperative hemoglobin less than 12.0 g/dL was defined as anemia. Due to our streamlined postoperative management plan, most patients underwent routine blood tests on the first and third day after surgery. Total estimated blood loss was measured by a formula described previously based on age, sex, height, weight, and hemoglobin [22,23].

Complications were identified from three sources: postoperative patient self-report questionnaires, inpatient hospital records, and surgeons’ outpatient office charts. Blood and platelet transfusions were also recorded. Incision complications included delayed incision healing, effusion, hematoma, superficial surgical site infection (SSI), and deep SSI. Deep vein thrombosis (DVT) and pulmonary embolism (PE) were confirmed by lower limb color Doppler ultrasound and computed tomography angiography of the pulmonary arteries in patients with symptoms within 90 days. Neurovascular injury, severe postoperative vomiting, high fever (temperature > 39 °C except for infection), myocardial infarction, and respiratory and urinary system infection within 30 days were recorded. Periprosthetic infections and periprosthetic fractures at any time were also recorded. Revision surgery was defined as replacing components for any reason. Reoperations and 90-day readmissions related to the knee joint and complications were also recorded.

### 2.3. Statistical Analysis

Statistical analyses were conducted using SPSS 23 (SPSS Inc., Chicago, IL, USA). Standard descriptive statistics were utilized for the reporting of all data including mean ± SD for continuous variables and count for categorical variables. A comparison of the categorical variables was conducted using the Fisher’s exact test. Continuous variables were compared using a two-sample *t*-test or Wilcoxon rank-sum test. Predictors that were found clinically or statistically significant in the univariate analysis were included in the multivariate analysis to estimate odds ratios and 95% confidence intervals of post-operative anemia. The final models for multivariate analysis were chosen using the enter selection method. A *p* value < 0.05 was taken as statistically significant.

## 3. Results

### 3.1. Demographic Characteristics

Thirty pSS patients in total were included, among which six patients underwent simultaneous bilateral TKAs; there were 36 limbs in total. After matching, 60 patients were included in the non-pSS group, and 12 patients underwent simultaneous bilateral surgery, resulting in 72 limbs. Patients in the two groups were followed up for at least one year, and the mean follow-up time was 6.84 ± 4.69 years and 6.76 ± 4.37 years, respectively (*p* > 0.05). All pSS patients were female, and no differences were observed in age, BMI, ASA scores, or aCCI between the two groups after matching (*p* > 0.05). There were five limbs of valgus deformities in the pSS group and four in the non-pSS group (*p* > 0.05). The two groups had no difference in operation time or length of hospital stay (*p* > 0.05). See Table 1 for details.

### 3.2. Knee Function and Health-Related Quality of Life Outcomes

Of the patients, 83.3% of the pSS group and 90% of the non-pSS group returned the patient-reported outcomes questionnaires. Patients in the pSS group had higher preoperative WOMAC pain scores and post-activity VAS scores than those in the non-pSS group, but there was no difference in stiffness, function, and total WOMAC score. At the last follow-up, both groups had improved in all components of the WOMAC score and total WOMAC score compared with preoperative scores, and there was no difference between the two groups (*p* > 0.05). There was no difference in VAS score (0.83 ± 0.65 vs. 0.72 ± 0.67) and FJS (75.53 ± 12.90 vs. 74.60 ± 11.48) between the two groups at the last follow-up (*p* > 0.05). In terms of quality of life, the EQ-VAS of pSS patients was worse than that of the control group before surgery (*p* < 0.05), but no difference was observed in other parts, and there was no difference in each component and EQ-VAS between the two groups after surgery. At the last follow-up, three patients in the pSS group and four patients in the non-pSS group needed walking aids (*p* > 0.05). The satisfaction of the two groups was 93.93 ± 4.66 points and 93.87 ± 3.63 points, respectively (*p* > 0.05). See Table 2 and Table 3 for details.

### 3.3. Hematological Outcomes

There was no significant difference in preoperative hemoglobin, albumin, plasma potassium, or platelet count (*p* > 0.05). The lowest postoperative hemoglobin was 104.10 ± 12.31 g/L in the pSS group and 111.75 ± 17.41 g/L in the non-pSS group (*p* < 0.05). The lowest postoperative albumin values in the two groups were 32.23 ± 4.12 g/L and 35.25 ± 4.57 (*p* < 0.05), respectively. No differences were observed in postoperative plasma potassium, platelet count, and total blood loss (*p* > 0.05). See Table 4 for details.

### 3.4. Perioperative Complications

There were 21 cases of anemia in the pSS group and 24 cases in the non-pSS group (*p* < 0.05). In terms of postoperative hypoalbuminemia and low platelet count, the pSS group had more patients than the osteoarthritis group (*p* < 0.05). Blood transfusion was performed in four patients in both groups (*p* > 0.05). Three patients received platelet transfusion in the pSS group, while none received platelet transfusion in the non-pSS group (*p* < 0.05). Among the three transfusions in the pSS group, two patients had preoperative platelet < 50×10^9^ /L, and one patient had postoperative platelet < 50×10^9^/L. There were two patients in the pSS group and one patient in the non-pSS group with pulmonary infections, of which the two patients in the pSS group were complicated with pulmonary interstitial fibrosis before surgery. These were all cured after antibiotic treatment. There was no significant difference in other complications, and no 90-day readmission, revision, or reoperation occurred in either group (*p* > 0.05). See Table 5 for details.

### 3.5. Risk Factors for Postoperative Anemia

Among the univariate factors, the predictors associated with postoperative anemia in pSS patients included extended surgery time (*p* = 0.047) and low preoperative Hb (*p* = 0.015). The multivariate analysis included significant predictors from the univariate analysis. Unilateral or bilateral procedures were forced into the mode as variables of interest. By multivariate analysis, low preoperative Hb was a statistically significant predictor of postoperative anemia. A 1 g/dL increase in hemoglobin was associated with decreased risk of postoperative anemia (OR = 0.334, 95% CI (0.125–0.889), *p* < 0.05). See Table 6 and Table 7 for details.

## 4. Discussion

As the number of patients with pSS continues to rise and common musculoskeletal complications such as end-stage arthritis continue to increase [5,6], it is important to evaluate whether this young and mostly immunosuppressed patient group can still achieve good results after TKA and to ensure that they do not face a higher risk of complications than the general population without rheumatic diseases. This research highlights that patients with pSS who underwent TKA experienced significantly improve knee function and quality of life despite having higher rates of anemia, hypoalbuminemia, and platelet transfusions, and there were no differences in other complications, readmissions, or reoperations. In addition, multivariate analysis showed that low preoperative Hb was a risk factor for postoperative anemia in pSS patients underwent TKA.

Previous studies showed that patients with pSS with joint symptoms had RF more often than those without joint symptoms (45% vs. 33%), and 5–10% of pSS patients were positive for anti-citrullinated peptide antibody (ACPA) [24,25]. pSS patients with RF or who are ACPA-positive have more severe inflammatory arthritis [26,27,28,29]. Our results were similar and revealed RF (+) in 55.6% of patients in this pSS cohort. Compared with patients without pSS, patients with pSS had higher preoperative WOMAC pain and post-activity VAS scores, but there were no differences in VAS pain scores between the groups at the last follow-up, and they were substantially improved. Similar improvements in functional outcomes have also been found in SLE and RA patients. In a follow-up study of 31 SLE patients undergoing TKA, Issa et al. found that SLE patients achieved similar Knee Society Scores (KSS) compared to non-SLE patients [11]. In addition, no differences were observed in the Short Form—36 physical or mental components after a mean six-year follow-up [11]. In the TKA study of 23 RA patients, Xu et al. found that the Hospital for Special Surgery (HSS) score and KSS for pain were satisfactorily improved, and the range of motion and quality of life of the knee were significantly improved [9]. However, some studies have found that the functional results of patients with RA after TKA were worse than those of patients with OA, but there was no difference in pain improvement and satisfaction [10]. In our case, although patients with pSS had more severe knee pain and EQ-VAS scores before surgery, they achieved similar postoperative functional and quality of life outcomes as patients without pSS.

In this study, patients with pSS had more postoperative anemia than those without pSS, and low preoperative Hb was an independent risk factor for postoperative anemia in pSS patients. Preoperative hemoglobin in pSS patients was lower than that in the non-pSS group, but this was not a statistically significant difference (*p* > 0.05). Normocytic normochromic anemia occurs in approximately 20% of patients with pSS and might have been caused by various factors [30]. The hematologic system is one of the most affected systems of pSS, mainly because there are many kinds of autoantibodies in the serum of pSS patients. In addition to ANA, anti-SSA, anti-SSB, and other specific autoantibodies, there are also anti-erythrocyte, anti-neutrophil, and anti-platelet autoantibodies, which can contribute to a decrease in hemoglobin [30,31]. In addition, long-term use of immunosuppressants can cause bone marrow suppression, resulting in hematopenia [32]. Some drugs can affect hematopoietic raw material levels, such as methotrexate and sulfapyridine, which reduce folic acid levels [33]. Low preoperative hemoglobin was also found to be a high-risk factor for transfusion after joint replacement in a study of RA patients [34]. Postoperative anemia will increase the number of days in the hospital, infection rate, and mortality. In addition, severe anemia will increase the patient’s fatigue and cause hypotension, which is not conducive to the patient’s early functional exercise and underground activities [35,36]. For patients with pSS, erythropoietin or ferralia should be given preoperatively to optimize hemoglobin levels, thus reducing the risk of postoperative anemia and even blood transfusion.

Patients with pSS may present with abnormal cell count in any cell line, and thrombocytopenia occurs in 5% to 13% of SS patients and may occur at any time during the disease [32]. Low platelets can increase intraoperative and postoperative bleeding and even increase the risk of disseminated intravascular coagulation (DIC) and hemorrhagic shock. In this study, there were six patients with preoperative thrombocytopenia, including two patients who required platelet infusion before surgery and one patient who required platelet infusion after surgery. No difference was observed in postoperative blood loss between the two groups. The main cause of pSS thrombocytopenia is the increase in peripheral blood destruction, which may be related to antiplatelet antibodies. Multiple autoantibodies adsorbing on the platelet surface can destroy the structure and integrity of the platelet membrane, and the clinical manifestation is peripheral blood thrombocytopenia [37]. On the other hand, patients in the pSS group had more postoperative hypoalbuminemia in comparison with those in the non-pSS group, and hypoalbuminemia is a key risk factor for poor incision healing and infection [38]. In this study, 86.7% of patients had a history of glucocorticoid use. Long-term use of glucocorticoids had adverse effects on albumin metabolism [39]. Previous studies have shown that there is a higher incidence of postoperative complications after TKA for inflammatory arthritis, but in our study, no difference was found between pSS patients and non-pSS patients in terms of incision complications and deep vein thrombosis, and no complications such as periprosthetic infection, 90-day readmission, or revision were found.

Our research had certain limitations, and our outcomes should be interpreted with these in mind. First, although we used cohort matching, there was still potential bias due to the inherent limitations of retrospective studies. Second, the sample size was small, which limited our ability to identify differences in rare complications and to conduct regression analyses to explore risk factors for complications. However, the incidence of pSS was lower compared to other rheumatic diseases such as SLE and RA, which means that the potential number of patients performing TKA was even lower. In addition, the data in this study were based on detailed hospital records and were more detailed than large database studies. Third, many surgeons have used different implants from different manufacturers. However, we believe that such a diverse setting can more truly reflect the setting in the real world and make the results more generalizable. Fourth, not all patients reported outcome questionnaires, introducing a possible source of bias when comparing clinical outcomes. Fifth, although all patients had similar joint function at the last follow-up and no reoperation was performed, there was a lack of comparison of radiographic data such as prosthesis loosening at the last follow-up.

## 5. Conclusions

Even with these shortcomings, we consider this study to be a significant addition to the literature given the limited data available for comparison with matched controls in this particular patient population. This study showed that patients with pSS improved their knee function and quality of life after TKA in this young, immunosuppressed patient population compared with the general population. Patients with pSS had a higher incidence of postoperative anemia and hypoalbuminemia and were more likely to require platelet transfusion during the perioperative period. However, there was no difference in other complications, readmission, and reoperation. Based on our results, rheumatologists and orthopedists should not be reluctant to perform TKA on pSS patients because of concerns about the outcome or potential medical complications, but they should be aware that perioperative hematologic management is critical since low hemoglobin is a risk factor for postoperative anemia. This information can also be used to consult patients with pSS who are considering knee arthroplasty. Prospective multi-center studies with large sample sizes are needed to evaluate the long-term effects of TKA in pSS patients.

## Figures and Tables

**Figure 1 jcm-11-07438-f001:**
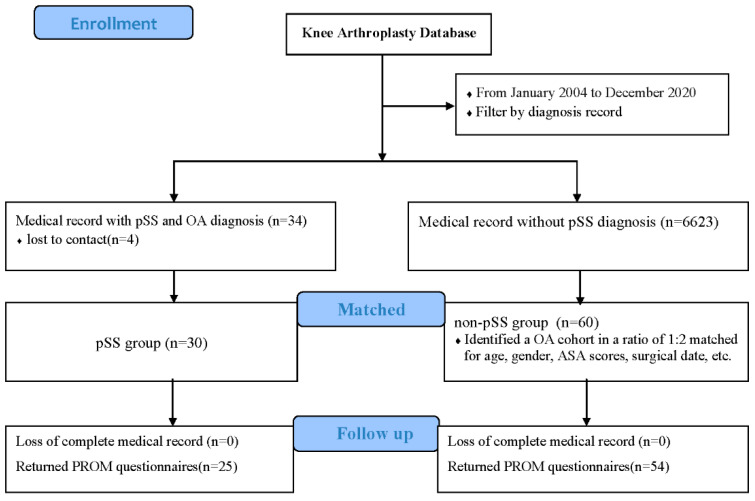
Flow chart. Abbreviations: pSS = primary Sjögren’s syndrome; OA = osteoarthritis; PROM = patient-reported outcome measure.

**Table 1 jcm-11-07438-t001:** Demographic characteristics.

	pSS Group	Non-pSS Group	*p* Value
Age (years)	63.33 ± 8.28	62.83 ± 8.93	0.798
Gender (female/male)	30/0	60/0	1.000
BMI (kg/m^2^)	24.15 ± 2.48	24.76 ± 2.20	0.241
ASA score	2.03 ± 0.56	2.12 ± 0.61	0.533
aCCI	2.63 ± 0.93	2.40 ± 0.96	0.275
Alignment (varus/valgus)	31/5	68/4	0.267
Operation time (min)	127.27 ± 27.66	122.55 ± 16.93	0.320
Length of hospital stay (days)	14.33 ± 8.72	12.82 ± 5.79	0.328
Follow up time (years)	6.84 ± 4.69	6.76 ± 4.37	0.934

Abbreviations: BMI = body mass index; ASA = American Society of Anesthesiologists; aCCI = age-adjusted Charlson Comorbidity Index.

**Table 2 jcm-11-07438-t002:** Knee function.

	pSS Group	Non-pSS Group	*p* Value
Preoperative			
WOMAC-pain	15.93 ± 2.43	13.63 ± 3.24	0.001
WOMAC-stiffness	6.03 ± 1.90	5.92 ± 1.75	0.773
WOMAC-physical function	44.83 ± 13.74	48.52 ± 9.08	0.132
WOMAC-total	66.80 ± 14.21	68.07 ± 9.25	0.612
VAS pain score	6.93 ± 1.36	5.08 ± 1.84	0.001
Postoperative			
WOMAC-pain	0.63 ± 0.72	0.43 ± 0.65	0.186
WOMAC-stiffness	0.57 ± 0.63	0.50 ± 0.70	0.661
WOMAC-physical function	3.30 ± 1.26	3.47 ± 1.55	0.611
WOMAC-total	4.50 ± 1.66	4.40 ± 1.88	0.805
VAS pain score	0.83 ± 0.65	0.72 ± 0.67	0.431
FJS	75.53 ± 12.90	74.60 ± 11.48	0.728

Abbreviations: WOMAC = The Western Ontario and McMaster Universities Osteoarthritis Index; VAS = visual analogue scale; FJS = forgotten joint score.

**Table 3 jcm-11-07438-t003:** Health-related quality of life assessment outcomes.

	pSS Group	Non-pSS Group	*p* Value
Preoperative			
EQ-mobility	3.23 ± 1.22	2.97 ± 1.19	0.323
EQ-self care	3.27 ± 1.53	3.08 ± 1.24	0.543
EQ-usual activity	3.77 ± 0.90	3.70 ± 0.87	0.735
EQ-pain	3.83 ± 0.79	3.75 ± 1.10	0.712
EQ-anxiety	2.90 ± 0.89	3.23 ± 1.25	0.196
EQ-scores	0.19 ± 0.21	0.18 ± 0.18	0.853
EQ-VAS	57.63 ± 8.89	62.37 ± 9.60	0.026
Postoperative			
EQ-mobility	2.00 ± 0.59	1.78 ± 0.64	0.124
EQ-self care	1.30 ± 0.47	1.22 ± 0.42	0.392
EQ-usual activity	1.40 ± 0.56	1.47 ± 0.54	0.586
EQ-pain	1.23 ± 0.50	1.33 ± 0.54	0.401
EQ-anxiety	1.53 ± 0.63	1.82 ± 0.85	0.111
EQ-scores	0.86 ± 0.080	0.85 ± 0.077	0.986
EQ-VAS	73.60 ± 12.49	77.43 ± 12.02	0.163
Walking aid (Y/N)	3/27	4/56	0.682
Patient satisfaction	93.93 ± 4.66	93.87 ± 3.63	0.941

Abbreviations: EQ = European Quality of Life Scale; VAS = visual analogue scale.

**Table 4 jcm-11-07438-t004:** Perioperative hematological index.

	pSS Group	Non-pSS Group	*p* Value
Preoperative			
Hemoglobin (g/dL)	12.01 ± 1.55	12.54 ± 1.53	0.128
Albumin (g/L)	36.37 ± 4.41	38.19 ± 4.42	0.067
Plasma potassium (mmol/L)	4.01 ± 0.51	4.18 ± 0.64	0.208
Platelet count (×10^9^/L)	189.43 ± 68.55	211.09 ± 65.16	0.148
Postoperative			
Hemoglobin (g/L)	104.10 ± 12.31	111.75 ± 17.41	0.034
Albumin (g/L)	32.23 ± 4.12	35.25 ± 4.57	0.003
Plasma potassium (mmol/L)	3.69 ± 0.42	3.83 ± 0.49	0.188
Platelet count (×10^9^/L)	157.63 ± 64.66	176.23 ± 51.42	0.139
Total blood loss (mL)	526.90 ± 201.02	532.07 ± 208.08	0.911

**Table 5 jcm-11-07438-t005:** Postoperative complications.

	pSS Group	Non-pSS Group	*p* Value
Anemia	21/9	24/36	0.013
Hypoalbuminemia	22/8	28/32	0.024
Hypokalemia	10/20	16/44	0.623
Low platelet	7/23	1/59	0.002
Blood transfusion	4/26	4/56	0.433
Platelet transfusion	3/27	0/60	0.035
Incision complications	2/28	2/58	0.598
Deep vein thrombosis	0/30	1/59	0.999
Pulmonary embolism	0/30	0/60	1.000
Myocardial infarction	0/30	0/60	1.000
Pulmonary infection	2/28	1/60	0.252
Urinary tract infection	2/28	3/57	0.999
High fever	5/25	7/53	0.525
Vomiting	4/26	6/54	0.726
Neurovascular events	0/30	0/60	1.000
Periprosthetic infection	0/30	0/60	1.000
Periprosthetic fracture	0/30	0/60	1.000
Aseptic Loosening	0/30	0/60	1.000
90-day Readmission	0/30	0/60	1.000
Reoperation	0/30	0/60	1.000
Revision	0/30	0/60	1.000

All data are represented by the number of cases. Values are Y/N unless otherwise specified.

**Table 6 jcm-11-07438-t006:** Univariate analysis for postoperative anemia in pSS patients.

	Anemia	Non-Anemia	*p* Value
Age (years)	62.71 ± 8.76	64.78 ± 7.29	0.541
Gender (female/male)	21/0	9/0	1.000
BMI (kg/m^2^)	24.06 ± 2.70	24.36 ± 1.97	0.771
ASA score	2.00 ± 0.63	2.11 ± 0.33	0.624
aCCI	2.62 ± 0.97	2.67 ± 0.87	0.900
Operation time (min)	133.76 ± 26.18	112.11 ± 26.26	0.047
Age at diagnosis (years)	55.48 ± 8.42	56.78 ± 6.53	0.683
pSS Duration (years)	7.24 ± 4.94	8.00 ± 4.66	0.697
Unilateral/bilateral	15/6	9/0	0.141
Renal involvement (Y/N)	2/19	0/9	0.999
Digestive involvement (Y/N)	1/20	1/8	0.517
Hematological involvement (Y/N)	7/14	3/6	0.999
Smoking (Y/N)	0/21	1/8	0.300
ESSPRI	6.62 ± 1.59	5.82 ± 1.93	0.246
Preop Hb (g/dL)	11.57 ± 1.51	13.03 ± 1.17	0.015
Albumin (g/L)	36.91 ± 4.78	35.11 ± 3.26	0.315
Platelet count (×10^9^/L)	176.62 ± 68.34	219.33 ± 62.62	0.119
Plasma potassium (mmol/L)	3.92 ± 0.53	4.22 ± 0.43	0.146
Anti-SSA (+) (Y/N)	8/4	5/7	0.414
Anti-SSB (+) (Y/N)	6/6	4/8	0.680
ANA (+) (Y/N)	13/3	9/2	0.999
RF (+) (Y/N)	9/7	6/5	0.999
Low C3 (Y/N)	5/8	4/10	0.695
Low C4 (Y/N)	4/9	4/10	0.999
High IgG levels (Y/N)	7/6	6/8	0.706
Glucocorticosteroids (Y/N)	19/2	7/2	0.563
“Stress-dose” steroids (Y/N)	16/5	6/3	0.667
Hydroxychloroquine (Y/N)	16/5	5/4	0.389
Total glucosides of paeony (Y/N)	13/8	6/3	0.999
bDMARDs (Y/N)	2/19	0/9	0.999
tsDMARDs (Y/N)	0/21	0/9	1.000

Abbreviations: BMI = body mass index; ASA = American Society of Anesthesiologists; aCCI= age-adjusted Charlson Comorbidity Index; pSS = primary Sjögren’s syndrome; ESSPRI = EULAR Sjögren’s Syndrome Patient Reported Index; Preop Hb = preoperative hemoglobin; ANA= antinuclear antibody; RF= rheumatoid factor; Low C3 = C3 < 0.73 g/L; Low C4 = C4 < 0.1 g/L; High IgG levels = IgG ≥ 17.00 g/L; bDMARDs = biological disease modifying anti-rheumatic drugs; tsDMARDs = target synthetic disease modifying anti-rheumatic drugs. + means a positive lab result.

**Table 7 jcm-11-07438-t007:** Multivariate Analysis for postoperative anemia in pSS patients.

	Odds Ratio (95% CI)	*p* Value
Preop Hb (g/dL)	0.334 (0.125–0.889)	0.028
Operation time (min)	1.058 (0.996–1.124)	0.065
Unilateral/bilateral	2.139 (0.069–66.68)	0.665

Abbreviations: Preop Hb = preoperative hemoglobin.

## Data Availability

The data associated with the paper are not publicly available but are available from the corresponding author upon reasonable request.

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
