# Peer review of "Total Knee Arthroplasty in Patients with Primary Sjögren’s Syndrome: A Retrospective Case-Control Study Matched Patients without Rheumatic Diseases"

_jcm, 2022, doi:10.3390/jcm11247438_

Round 1
Reviewer 1 Report
1. the authors describe the variables by which they matched cases to controls. were there only 60 controls with these criteria or more patients? if so, how were the 60 selected? randomly?
2. The authors perform a power calculation for anemia, which is only one of the study variables. No power calculation was made for other outcome variables and, surely, it does not make much sense since all cases with S. Sjogren's disease were included. This section should probably be deleted
3. Table 6 (univariate analysis of anemia-related predictors) includes a large number of variables, many of which have no apparent relationship to anemia (nervous system involvement, lung involvement, WOMAC scores, VAS pain score, ....). Should be reviewed. Not all available variables should be systematically analyzed, but only those with a clinical meaning.
Reviewer 2 Report
This is a very interesting research, which focused on the differences in peri-operative complications between pSS patients and OA patients without rheumatic diseases. They found that there were no significant differences in postoperative pain, knee function and health-related quality of life, as well as some other complications, the rates of 90-day readmission, or overall revision. But, pSS group had more patients with post-operative anaemia and hypoalbuminemia, and more patients need platelet transfusion. At the same time, authors figured out that the risk factor of post-operative anaemia in pSS patients was lower preoperative hemoglobin.
However, there are some issues which need further elaboration.
1. In the abstract, although the risk factor of post-operative anaemia in pSS patients was lower preoperative hemoglobin, these results are insufficient to conclude “Improving preoperative Hb level can reduce the risk of post-operative anaemia”.
2. Please provide the detail diagnostic criteria for pSS ,OA and anemia.
3. It is better to have a flow chart to illustrate the process of screening patients from the database.
4. Please provide the data of pellet count in Table 4.
5. In Table 6, the “pellet count” and “time from diagnosis of pSS to surgery” may could be do the univariate analysis.
6. Is there any influence of time and dose of using medication,such as glucocorticoids and disease modifying anti-rheumatic drugs, on the peri-operative complications?
7. Is there any possible to do the subgroup analysis of unilateral and bilateral?
8. “Operation time (min)” and “Unilateral/bilateral” may not be independent risk factors.
9. There are some spelling mistakes, such as at line 208.
